# Transient Expression of Flavivirus Structural Proteins in *Nicotiana benthamiana*

**DOI:** 10.3390/vaccines10101667

**Published:** 2022-10-06

**Authors:** Naveed Asghar, Wessam Melik, Katrine M. Paulsen, Bendikte N. Pedersen, Erik G. Bø-Granquist, Rose Vikse, Snorre Stuen, Sören Andersson, Åke Strid, Åshild K. Andreassen, Magnus Johansson

**Affiliations:** 1School of Medical Sciences, Inflammatory Response and Infection Susceptibility Centre (iRiSC), Faculty of Medicine and Health, Örebro University, SE-70362 Örebro, Sweden; 2Department of Virology, Division for Infection Control, Norwegian Institute of Public Health, N-0213 Oslo, Norway; 3Department of Production Animal Clinical Sciences, Norwegian University of Life Sciences, N-4325 Sandnes, Norway; 4Department of Public Health Analysis and Data Management, Public Health Agency of Sweden, SE-17165 Solna, Sweden; 5School of Science and Technology, The Life Science Centre, Örebro University, SE-70281 Örebro, Sweden

**Keywords:** VLP, tick-borne encephalitis virus, plant, tobacco, mice, flavivirus, vaccine, toxicity, protein expression, immunization

## Abstract

Flaviviruses are a threat to public health and can cause major disease outbreaks. Tick-borne encephalitis (TBE) is caused by a flavivirus, and it is one of the most important causes of viral encephalitis in Europe and is on the rise in Sweden. As there is no antiviral treatment available, vaccination remains the best protective measure against TBE. Currently available TBE vaccines are based on formalin-inactivated virus produced in cell culture. These vaccines must be delivered by intramuscular injection, have a burdensome immunization schedule, and may exhibit vaccine failure in certain populations. This project aimed to develop an edible TBE vaccine to trigger a stronger immune response through oral delivery of viral antigens to mucosal surfaces. We demonstrated successful expression and post-translational processing of flavivirus structural proteins which then self-assembled to form virus-like particles in *Nicotiana benthamiana*. We performed oral toxicity tests in mice using various plant species as potential bioreactors and evaluated the immunogenicity of the resulting edible vaccine candidate. Mice immunized with the edible vaccine candidate did not survive challenge with TBE virus. Interestingly, immunization of female mice with a commercial TBE vaccine can protect their offspring against TBE virus infection.

## 1. Introduction

Flaviviruses like yellow fever virus (YFV), dengue virus (DENV), tick-borne encephalitis virus (TBEV), West Nile virus (WNV), Japanese encephalitis virus (JEV), and Zika virus (ZIKV) are a threat to public health and have recently caused several disease outbreaks [1,2]. In particular, TBE has been classified as a notifiable disease in Sweden since July 2004, and the number of TBE cases has increased over the years [3]. According to Sweden’s public health agency, a record high number of TBE cases (n = 533) were reported in 2021 [4]. As there is no antiviral treatment available, vaccination remains the best protective measure against TBE. However, current TBE vaccines have limitations as they must be delivered by intramuscular injection, have a burdensome immunization schedule, and may fail due to poor immunogenicity in certain populations [5].

One promising strategy that could help improve flavivirus vaccines in general, and TBE vaccines in particular, is molecular farming, which is the production of pharmaceutically important recombinant proteins in plants. Molecular farming has several advantages over other eukaryotic protein production systems, including simplicity, speed, scalability, safety, and sustainability [6,7,8]. Compared to molecular farming, classical protein production in prokaryotic or other eukaryotic cells is expensive, laborious, and risks contamination with endotoxins or pathogens; furthermore, there are several constraints on the scaling-up process. Molecular farming has successfully been used for large-scale production in a short time and has been approved for production of several pharmaceutical proteins and vaccines [9,10,11,12,13]. Agroinfiltration is the commonly used technique in molecular farming that involves *Agrobacterium tumefaciens*-mediated transformation and production of recombinant proteins in plant cells for several days [14,15,16]. This transient protein expression is accomplished when *A. tumefaciens* transfers a specific part of its Ti (tumor-inducing) plasmid for integration into the genome of infected plant cells [17]. The flexibility and high speed of plant-based expression systems make them especially suitable for vaccine development in the management of outbreaks, epidemics, and pandemics by offering a prompt response though rapid production of subunit vaccines [18]. For example, molecular farming-based vaccines against influenza and SARS CoV-2 viruses have been approved for human use in Canada [19,20,21].

In addition to producing purified pharmaceutical proteins, molecular farming can also be used to produce edible transgenic plants to potentially serve as oral, edible vaccines. Edible vaccines are as safe as conventional virus vaccines and they do not require extensive processing, expensive purification, cold-chain transport, or sterile delivery by trained professionals [9]. The idea behind oral vaccines is that the antigens will be protected from gastric acid and enzymes by the plant cell walls, and can subsequently be delivered to the mucosal surfaces of the gastrointestinal tract to evoke an efficient mucosal immune response [22].

Virus-like particles (VLPs) are structures formed by the self-assembly of viral structural proteins. The VLPs mimic virus morphology but lack infectious genomic material [23], and their small size and the repetitive epitope patterns on their surfaces trigger a strong immune response [24]. These qualities make VLPs strong candidates for vaccine development, and it has been shown that a plant-based quadrivalent VLP influenza vaccine provides better protection than egg-based vaccines in humans [25]. A wide range of plant-based VLPs (including hepatitis B virus, influenza virus, human- and bovine-papillomavirus, bluetongue virus, norovirus, and cowpea mosaic virus) have been shown to elicit protective immune responses in animal models, and have been evaluated as vaccine candidates or for novel delivery systems [26]. In addition, VLPs for several flaviviruses, including TBEV, have been produced by solitary expression of the structural proteins in various hosts and evaluated as potential vaccine candidates [27].

The objective of this study was to explore transient expression of TBEV structural proteins in plants for developing plant-based edible vaccines in which plant leaves expressing the TBEV VLPs would be consumed directly. In addition, we evaluated whether the TBEV antibodies could be vertically transmitted from immunized female mice to offspring and provide protection against TBEV infection.

## 2. Materials and Methods

### 2.1. Recombinant Binary Vectors

A pEAQ-CME vector encoding TBEV structural proteins was constructed using TBEV C-prM-E sequence (accession number DQ401140.3). To increase expression, the TBEV genomic sequence was codon-optimized for the model plant *Nicotiana tabacum* (a close relative of one of our experimental plants), chemically synthesized, and cloned into a plant expression vector pEAQ-HT using NruI and XhoI restriction sites. A Kozak consensus sequence GCCACC was incorporated upstream of the TBEV start codon. A pEAQ-GFP vector was constructed using pEAQ vector [28] and a pJL3:p19 vector with p19 gene of tomato bushy stunt virus (TBSV) [29] was kindly provided by Mike Boehm.

### 2.2. Transformation

The recombinant vectors were transformed into competent *A. tumefaciens* strain LBA4404 using standard procedures and one minute freezing in liquid nitrogen [30]. The transformants were selected on LB plates with antibiotics (kanamycin 50 µg/mL, rifampicin 25 µg/mL and streptomycin 50 µg/mL). Plates were incubated at 28 °C for 2–3 days or until colonies appeared. Single colonies of *A. tumefaciens* transformed with a binary vector were separately inoculated into 3 mL LB media with antibiotics (kanamycin 50 µg/mL, rifampicin 25 µg/mL, streptomycin 50 µg/mL) and incubated at 28 °C for 2 days at 220 rpm. The culture was pelleted by centrifugation and re-suspended in 100 µL ddH2O. The resuspension was boiled for 5 min and centrifuged at 5000× *g* for 1 min. The supernatant was used as PCR template to confirm transformants containing pEAQ-CME vector. Colony PCR was performed using the forward primer 5′-TCTCTACTTCTGCTTGACGAGG-3′, the reverse primer 5′-AAGCTTGATATCGAATTCCCGG-3′ and Phusion high-fidelity DNA polymerase (New England BioLabs). The resulting PCR products were analyzed by electrophoresis.

### 2.3. Plant Growth

The seeds of 5 different species were germinated and grown in a growth room with a 12 h light/dark cycle. Plants were agroinfiltrated at specified weeks post germination to maximize transient protein expression: a close relative of commercial tobacco (*Nicotiana benthamiana*), 6 weeks; New Zealand spinach (*Tetragonia tetragonioides*), 9 weeks; lettuce (*Lactuca sativa*), 5 weeks; Swiss chard (*Beta vulgaris*), 7 weeks; spinach (*Spinacia oleracea*), 5 weeks. Plants were kept in sealed cultivation chambers with artificial light following agroinfiltration. The *N. benthamiana* used in this study is classified as a low or non-converter of alkaloids [30].

### 2.4. Syringe Infiltration

*A. tumefaciens* transformed with pEAQ-CME, pEAQ-GFP or pJL3:p19 were grown separately to an OD_600_ of 0.8–1.0 in LB medium supplemented with kanamycin 50 µg/mL, rifampicin 25 µg/mL and streptomycin 50 µg/mL at 28 °C with shaking (225 rpm). The bacterial cultures were mixed using 35 mL of *A. tumefaciens*/pEAQ-CME or *A. tumefaciens*/pEAQ-GFP and 15 mL of *A. tumefaciens*/pJL3:p19 and pelleted by centrifugation at 5000× *g* for 10 min. The pellet was resuspended in freshly made MMA solution infiltration buffer (10 mM 2-N-morpholinoethanesulfonic acid pH 5.6, 10 mM MgCl2, and 100 µM acetosyringone) and stir-incubated at room temperature in the dark for 2–4 h. Mixed cultures of *A. tumefaciens* were infiltrated into the abaxial surface of each leaf with 1 mL needleless syringe applying gentle pressure to minimize leaf damage.

### 2.5. GFP Expression

Plants were moved to a dark room and exposed to a hand-held UV lamp emitting 366 nm long wave UV light to visualize GFP expression in leaves. Photos were taken without a flash to visualize green fluorescence 7 days post infiltration (dpi).

### 2.6. Immunoblotting

Plant leaves were harvested 7 dpi and ground in liquid nitrogen to a fine powder using a pestle and mortar. Protein extraction was performed in RIPA buffer prior to boiling in 1xLDS sample buffer (Invitrogen). Proteins were separated by SDS-PAGE using precast 4–12% Bis-Tris gels (Invitrogen) and MES running buffer (Invitrogen). The proteins were transferred to nitrocellulose membranes using iBlot 2 Gel Transfer Device (Invitrogen) for detection with an in-house rabbit anti-TBEV E polyclonal antibody (1:200) and a mouse monoclonal anti-LGTV E 11H12 antibody (1:2000) (United States Army Medical Research, Institute of Infectious Diseases, Fort Detrick, Frederick, MD, USA).

### 2.7. Transmission Electron Microscopy

The samples were negative stained using 3 μl of the sample applied on glow-discharged carbon-coated and formvar-stabilized 400 mesh copper grids (Ted Pella) and incubated for approximately 30s. Excess of sample was blotted off and the grid was washed with MilliQ water prior to negative staining using 2% uranyl acetate. TEM imaging was done using HT7700 (Hitachi High-technologies) transmission electron microscope operated at 100 kV and equipped with a 2kx2k Veleta CCD camera (Olympus Soft Imaging System).

### 2.8. Animal Experiments

BALB/cAnNRj 5–6 weeks old mice were housed at the Norwegian Institute of Public Health, Oslo, Norway. Mice were originally purchased from JANVIER LABS (Le Genest-Saint-Isle, France) and acclimatized for one week. They were housed in plastic cages in a room with a 12 h light/dark cycle and controlled humidity (55 ± 5%) and temperature (20–24 °C). All mice were fed *ad libitum* with a standard maintenance diet from Special Diets Services (Witham, UK).

To test the oral toxicity of different plant species, a total of 70 female mice were housed in groups of five and fed diets supplemented with different plants for 48 h. *N. benthamiana*, *L. sativa*, and *B. vulgaris* were given as 25 g, 50 g, and 100 g supplements, while *S. oleracea* was given as 25 g and 50 g supplements, and *T. tetragonioides* was given as a 25 g supplement. The control mice were fed the standard maintenance diet. The mice were weighed on day 0 and on day 28 (at termination of the trial), and they were euthanized by cervical dislocation. All organs were examined for toxicity by gross pathological examination.

To test immunization, a total of 40 female and 5 male mice were housed in groups of five. The mice were immunized by feeding with 25 g of *N. benthamiana* leaves expressing TBEV VLPs (n = 10) or with 0.5 mL subcutaneous injections of Ticovac^TM^ (Pfizer, New York, NY, USA) (n = 10) at weeks 0, 2, and 4. To control for the effect of the plant, mice (n = 10) were fed 25 g PBS-infiltrated *N. benthamiana* leaves whereas control mice (n = 15) were fed the standard maintenance diet. Blood samples were taken from each mouse before each immunization. After week 5, half the mice (n = 5) from each treatment and control group were challenged with 10^6^ PFU of TBEV strain Hochosterwitz in 100 µL cell culture media by subcutaneous injection above the right foreleg. Control mice for TBEV challenge were injected with 100 µL of cell culture media. The mice were monitored for pathological symptoms and survival time. The remaining female mice (n=5) from each treatment and control group were mated with untreated males, and offspring were infected with the same batch of TBEV that had been used to infect the maternal mice. The mice were monitored and euthanized by cervical dislocation and all organs were examined for toxicity by gross pathological examination.

## 3. Results

### 3.1. Transient Protein Expression in N. benthamiana after Agroinfiltration

To study transient protein expression by agroinfiltration in *N. benthamiana* (tobacco) and *T. tetragonioides* (New Zealand spinach), cultures of *A. tumefaciens* transformed with pEAQ-GFP and pJL3:p19 were mixed and infiltrated. The pJL3:p19 was co-infiltrated to express p19 protein, a suppressor of gene silencing, to enhance expression of GFP by pEAQ-GFP vector. *N. benthamiana* leaves emitted strong green fluorescence 7 dpi reflecting successful GFP expression, whereas *T. tetragonioides* leaves showed almost no fluorescence (Figure 1B,C). We repeated the syringe agroinfiltration three times, but *T. tetragonioides* failed to express GFP. In fact, the bacterial suspension could not spread through *T. tetragonioides* in the same way that it did in *N. benthamiana*, and damage to the leaf structure was observed when we increased the infiltration pressure.

### 3.2. Successful Expression and Processing of TBEV Structural Proteins in N. benthamiana

Successful expression of flavivirus structural proteins requires post-translational cleavage by virus and host proteases. To investigate if TBEV structural proteins can be successfully expressed in tobacco, cultures of *A. tumefaciens* transformed with pEAQ-CME and pJL3:p19 were infiltrated into *N. benthamiana*. A band of ~50 kDa corresponding to TBEV E protein was visualized by both polyclonal and monoclonal antibodies against flavivirus E protein (Figure 2A,B). The transmission electron microscopy of pEAQ-CME-infiltrated *N. benthamiana* leaves showed circular particles of ~50 nm diameter (Figure 2C) which may have been formed by post-translational processing of C-prM-E polyprotein and self-assembled into VLPs.

### 3.3. Orally Administrated N. benthamiana Was Tolerated in Mice

Although *N. benthamiana* showed high GFP expression and successful production of TBEV VLPs, it may have limitations as a vector for oral immunization due to the risk of toxicity conferred by alkaloids. To study the toxicity of the various plant species in mice, we fed them a diet supplemented with leaves from five different plant species (*N. benthamiana*, *T. tetragonioides*, *L. sativa*, *B. vulgaris*, and *S. oleracea*). The edible plant species (all other than *N. benthamiana*) were used as controls to test these species for future development of oral vaccines. All plant species were accepted by the mice and eaten in comparable amounts (consumption of *N. benthamiana* was slightly reduced for the two highest amounts of leaf material). No general effect of these supplements on animal weight was observed except for a slightly lower weight gain for the mice that were fed the diet supplemented with *S. oleracea* or the higher doses of *N. benthamiana* (50 g and 100 g) (Table 1).

The acute toxicology study was conducted according to OECD 420 guidelines. The animals in control group 2 gained substantially less weight than the animals in control group 1 and one animal was removed from calculation of the average because it did not show any weight gain after 28 days. Weight gain of all animals in control groups was 1.3–3.0 g. The plant leaves were readily consumed by the mice and moisture loss in the plants was not taken into the calculation, since the observed loss was <0.36%. All animals appeared healthy throughout the study, and they showed healthy organs in the gross pathological examination by the veterinarian.

### 3.4. Immunization with Commercial TBEV Vaccine Protected the Offspring

We were interested in testing the immunogenicity of our edible vaccine candidate and comparing it to the commercial TBEV vaccine, Ticovac. We immunized the animals by feeding them fresh *N. benthamiana* leaves expressing TBEV VLPs or with Ticovac injection. After 5 weeks, 50% of the immunized and control animals were infected with TBEV. All animals immunized with edible vaccine candidates and in control groups died 6–8 days after TBEV infection, while the mice that received Ticovac survived the challenge (Table 2). The gross pathological examination showed that the animals fed with *N. benthamiana* leaves had comparatively smaller organs than the other animals. All the mice that died after TBEV challenge showed massive bleeding in the stomach and upper part of the duodenum.

We also wanted to test if the mice immunized with the developed vaccine candidate and/or Ticovac could transmit sufficient antibodies to offspring to protect them against TBEV infection. For this experiment, the remaining 50% of the immunized and control animals were mated with untreated males. Their offspring were challenged with TBEV on day 21 post-birth and monitored for pathological symptoms and survival time. The offspring mice that had been vertically immunized (that is, their mothers had been vaccinated with Ticovac) remained healthy after TBEV challenge, while offspring from the other groups died 6–8 days post TBEV infection (Table 2).

## 4. Discussion

In this study, we used transient expression of TBEV C-prM-E in *N. benthamiana* (a tobacco relative) as a first step in an attempt to develop an edible TBE vaccine. We evaluated the oral toxicity of various plant species and immunogenicity of the edible vaccine candidate in mice. Flavivirus VLPs are commonly produced by expressing a polyprotein comprising C-prM-E or prM-E that is post-translationally cleaved into individual proteins that self-assemble to produce mature VLPs [27]. In this study, we used a pEAQ-CME vector to express C, prM, and E of TBEV as a single polyprotein. The pEAQ vectors are well characterized for VLP production in plants [31,32]. Flavivirus prM protein stabilizes the E protein, and molecular interaction between prM and E is important for flavivirus VLP production [33]. Flavivirus E protein has an indispensable role in cellular binding and membrane fusion between virus and host cell [34,35]. E protein is one of the most potent flavivirus antigens because it contains several epitopes actively recognized by the host immune system and it is a primary target of neutralizing antibodies [36]. Analysis of pEAQ-CME-derived expression of TBEV structural proteins in *N. benthamiana* by immunoblotting showed a band of ~50 kDa when stained with flavivirus anti-E antibodies. This observation indicates that *N. benthamiana* can provide a suitable environment, similar to mammalian cells, for post-translational processing of the polyprotein to produce correctly cleaved E protein. Similar observations showing ER insertion and post-translational processing of HBsAg in plants have been reported previously [37]. In addition, atomic resolution cryo-electron microscopy has demonstrated that the VLPs of cowpea mosaic virus (CPMV) produced in *N. benthamiana* are identical to native virus particles [38].

The choice of viral vectors used to express foreign proteins depends on the plant host and the size of the gene to be expressed so that concomitant protein folding is not disturbed. Binary vectors derived from plants viruses, such as tobacco mosaic virus (TMV), tobacco rattle virus (TRV), CPMV, and potato virus X (PVX) have been used to express recombinant foreign proteins in plants [31,39], and transient expression of vaccine antigens in tobacco after *Agrobacterium*-mediated transformation with TMV- and CPMV-based viral vectors has previously been shown [40,41]. In this study, we used a CPMV-based vector to express TBEV structural proteins in *N. benthamiana* by agroinfiltration. In addition, we used a second vector encoding the gene-silencing suppressor p19 from TBSV because p19 has been demonstrated to have enhanced transient expression in plants [42]. We used sequences of TBEV structural genes that had been codon-optimized for expression in *N. tabacum* to achieve higher expression levels of recombinant proteins in *N. benthamiana* [41,42,43]. Indeed, we achieved high protein expressions at 7 dpi, which is consistent with previous studies [29,41].

An ideal candidate plant for edible vaccine development would have high expression of recombinant proteins and non-toxic traits. In this study, we used *N. benthamiana* and *T. tetragonioides* (New Zealand spinach) for recombinant protein expression, and the former showed much stronger expression than the latter. *T. tetragonioides* was used as a putative edible transgenic plant system expressing a VLP vaccine candidate that could be consumed for immunization. The two factors that might have hindered agroinfiltration and subsequent transient protein expression in *T. tetragonioides* are its leaf morphology (in terms of thickness, cell composition, or density) and the fact that the expression system was optimized for *N. tabacum*. The other plant we investigated, *N. benthamiana*, is a versatile model for replication of plant viruses and expression of recombinant protein using virus-derived expression vectors [44]. However, this non-edible tobacco relative is not particularly attractive as an edible vaccine platform due to the toxic alkaloids in its leaves, primarily the addictive and toxic nicotine, but also smaller amounts of nornicotine (a suspected carcinogen [45]), anatabine, and anabasine [46]. The *N. benthamiana* used in the present study is classified as a low or non-converter of alkaloids. In the toxicity experiment, we did not observe any visible gross pathological changes in the animals except a slightly lower weight gain in animals that were fed with higher doses of *N. benthamiana* (50 g and 100 g). Nevertheless, we chose to proceed with low doses (25 g) in the immunization experiment. The mice were fed with fresh *N. benthamiana* leaves because curing leaves is known to elevate the proportion of nornicotine to nicotine in the leaves [46]. Despite these measures, the immunization experiments showed that the mice fed with *N. benthamiana* had smaller organs than the other mice, and it has been previously shown that tobacco can cause significant reduction in body and organ weights in mice and rats [47,48].

Despite their potential advantages, edible vaccines face the problem of dose quantification and quality control in the raw plants without purification [49]. In the immunization experiment in which mice were fed with *N. benthamiana* leaves expressing TBEV structural proteins, we did not observe any noticeable immune response. The lack of adequate immune responses after oral immunization could be due to several factors, including problems with stability in the gastrointestinal tract, dosage, protein folding or modifications, a need for repeated exposure, and lack of adjuvant. It has been reported that feeding mice with raw plants requires an adjuvant or priming with purified VLPs to trigger an optimal immune response [37]. The immunogenicity of the current vaccine may be improved by using an appropriate adjuvant [50], which, however, requires additional studies.

Flaviviruses such as JEV, TBEV, and WNV mainly cause encephalitis [51] while other flaviviruses like YFV, DENV, Omsk haemorrhagic fever virus, and Kyasanur Forest disease virus can cause haemorrhage [52,53]. We observed massive bleeding in the stomachs and upper parts of the duodenum in all the mice that died after infection with TBEV strain Hochosterwitz. These rare but important observations were supported by the clinical findings of TBEV and JEV infections in Russia and India [54,55].

Maternal immunization, which induces virus-specific antibodies that are subsequently dispensed to the offspring through placenta or breastfeeding, remains an effective public health strategy against several virus infections. In this study, female mice vaccinated with Ticovac and their offspring showed a 100% survival rate after TBEV challenge, and none of the infected animals developed any clinical symptoms (this example of maternal immunization in mice corroborates a recently published work based on another live-attenuated flavivirus vaccine [56]). The observations of this study are based on an edible TBEV vaccine candidate and immunization in mice, and extrapolations to other viruses and humans should not be made at this time.

In conclusion, we have shown that TBEV structural proteins can be expressed in *N. benthamiana* and self-assembled to form ~50 nm particles similar to TBEV virions. An oral toxicity experiment showed that *N. benthamiana* is tolerated in mice. However, immunization with the edible vaccine candidate could not protect the mice against TBEV challenge. The commercially-available TBE vaccine Ticovac provided complete protection to both female mice and their offspring.

## Figures and Tables

**Figure 1 vaccines-10-01667-f001:**
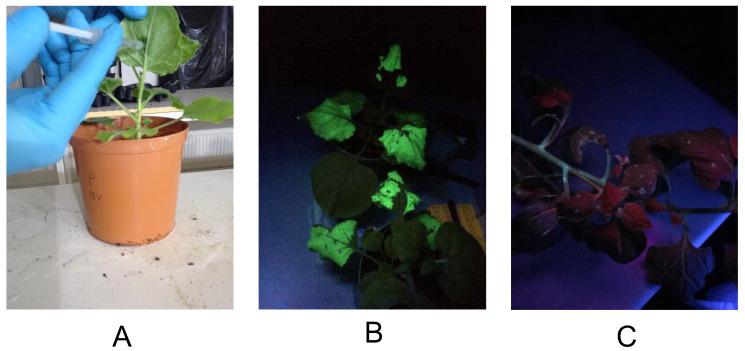
GFP expression in *N. benthamiana* and *T. tetragonioides* after agroinfiltration. (**A**) Syringe-infiltration of plant leaves with a mixture of *A. tumefaciens* cultures (OD_600_ 0.8–1.0) containing pEAQ-GFP and pJL3:p19 binary vectors. (**B**) *N. benthamiana* leaves showing strong GFP fluorescence. (**C**) *T. tetragonioides* leaves showing very weak GFP fluorescence.

**Figure 2 vaccines-10-01667-f002:**
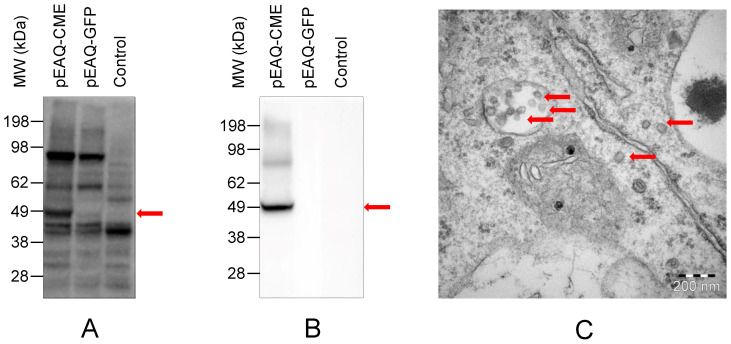
Expression and assembly of TBEV structural proteins producing virus like particles (VLP) in *N. benthamiana*. Plant leaves infiltrated with the transformed *A. tumefaciens* (OD_600_ 0.8–1.0) were analyzed 7 days post infiltration. Protein E expression by pEAQ-CME in *N. benthamiana* was visualized by Western blotting using polyclonal (**A**) and monoclonal (**B**) antibodies against the E protein (red arrow). Protein extracts from pEAQ-GFP and non-infiltrated (control) leaves were used as negative controls (**A**,**B**). Particles similar to TBEV VLPs, ~50 nm in diameter, (red arrows) localized in the endosome and other parts as visualized by transmission electron microscopy (**C**).

**Table 1 vaccines-10-01667-t001:** Weight gain of BALB/cAnNRj mice after feeding with plant-supplemented diets for 48 h.

Plant Specie	Plant/Cage (g)	Plant Consumed (g)	Average Weight Gain (g ± SD)
*Beta vulgaris*	25	23.0	2.56 ± 0.43
*Beta vulgaris*	50	44.7	2.42 ± 0.45
*Beta vulgaris*	100	74.5	2.58 ± 0.15
*Nicotiana benthamiana*	25	22.2	2.46 ± 0.67
*Nicotiana benthamiana*	50	39.5	1.58 ± 0.63
*Nicotiana benthamiana*	100	67.3	1.70 ± 1.22
*Lactuca sativa*	25	23.7	2.38 ± 1.03
*Lactuca sativa*	50	46.6	2.46 ± 0.39
*Lactuca sativa*	100	73.9	2.74 ± 1.12
*Tetragonia tetragonioides*	25	23.2	2.36 ± 0.88
*Spinacia oleracea*	25	23.7	1.54 ± 0.56
*Spinacia oleracea*	50	44.1	1.70 ± 0.30
Control 1			2.62 ± 0.25
Control 2			1.95 ± 0.66

**Table 2 vaccines-10-01667-t002:** Immunization schedule followed by infection with TBEV strain Hochosterwitz in BALB/cAnNRj mice. Mice were housed in groups of five. Female mice (n = 40) were used for immunization and 5 male mice were used for breeding. Mice survived (green) or died (red) post TBEV infection.

Treatment	*N. benthamiana*+VLP		*N. benthamiana*		Ticovac		Control Female Mice		Control Male Mice
First immunization (day 0)	25 g	25 g	25 g	25 g	0.5 mL	0.5 mL			
Second immunization (day 14)	25 g	25 g	25 g	25 g	0.5 mL	0.5 mL			
Third immunization (day 28)	25 g	25 g	25 g	25 g	0.5 mL	0.5 mL			
TBEV infection of adult mice (day 35)	TBEV	Breeding	TBEV	Breeding	TBEV	Breeding	TBEV	Breeding	Breeding
TBEV infection of offspring (day 21 after birth)		TBEV		TBEV		TBEV		TBEV	

## Data Availability

Not applicable.

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
