# Peer review of "Transient Expression of Flavivirus Structural Proteins in Nicotiana benthamiana"

_vaccines, 2022, doi:10.3390/vaccines10101667_

Round 1
Reviewer 1 Report
Dear authors of the manuscript Vaccines-1914475 Transient expression flavivirus structural proteins in Nicotiana benthamiana, I have been reading your manuscript with great interest and have the following comments to make:
General: The findings are of importance to further development of an edible TBE-vaccine. The presentation of the findings needs, however, a higher level of structure and clarity. It might be a good idea to split this manuscript into two separate? The English language should be improved.
Major comments:
Abstract: The abstract should better reflect the study in terms of the amount of text devoted to background, method, and main results. Eg the fact that the mice who were given the oral vaccine candidate died when challenged with TBE-virus should be mentioned. Please avoid the expression that "TBE is one of the most dangerous diseases" and use instead a medical term/sentence for how TBE affects people. For reference 4 this is a link to a Swedish webbsite, in Swedish. For increased transparency refer to a peer-reviewed paper in English, regarding TBE-vaccine failures in Sweden or elsewhere.
Methods and Results: Please improve structure so that methodology is described only under "Methods" and not under "Results" or "Discussion"
Discussion: No new results are to be found in this section. Describe shortly you main findings followed by a discussion with reference to other relevant published data as well as Strenghts and Limitations. Subheadings could be used for increased clarity. Be aware that this is a study of both expression of flaviviruses in plants and the effect of edible plants/vaccine in mice and extrapolations to humans are not appropriate. The finding of vertical transmission of TBEV antibodies in the off-spring after vaccination of the pregnant mice with a commercial vaccine are not really relevant for the study objective and experiments done. It is known that IgG from natural infection or vaccination can be transferred vertically in humans, but maternal vaccination is not applicable since newborn humans are not a risk group for TBE-infections.
Reviewer 2 Report
This is a very interesting study which shows the future of vaccines.
The study protocol is well designed and described in the manuscript.
line 22 „the most dangerous diseases”
I do not think any disease can be dangerous. It can by deadly or cause severe complications. Most cases of TBE are asymptomatic or mild. I am aware the disease can be life-threatening too.
line 37 “expensive” this term is VERY relative
line 51
“TBEV presents a significant threat in Europe and particularly in Eastern Europe. For instance, TBE has been notifiable in Sweden......”
This sentence suggests that Sweden is located in Eastern Europe. I believe it is not.
line 55
“Despite considerable advantages, current vaccines have limitations as they are invasive*, expensive*, have a burdensome immunization schedule*, and may display vaccine failure due to poor immunogenicity in the elderly.”
What does the “invasive vaccine” mean? To my knowledge there are only invasive diseases e.g. invasive meningococcal disease, invasive pneumococcal disease.
* provide citation if authors think this statement is true. Truly TBE vaccine requires couple injectable doses for primary vaccination and boosters as well.
line 285
The objective of the study appears for the first time in Discussion section of the manuscript.
Objectives should appear in the introduction of the research paper at the end of the problem statement.
In opposite authors should consider moving lines 102-105 from Introduction to Discussion
A summarized conclusion from the study at the end of the manuscript would be appreciated.
Round 2
Reviewer 1 Report
Dear authors, Thank You for considering my comments and appropriate changes have been made. Please, however, omit reference to/text about human/maternal TBE/immunization in the Discussion part. This is not relevant in a human setting, maternal IgG will disappear from the infants blood within 9 months and the TBE-risk for an infant is practically not existing. Your comments about this might inadevertenly cause confusion about human maternalTBE/immunization. It must be clear that your Discussion is about findings in mice and why do you think this is important (future knowledge for other experiments etc).
